

# Observing the role of wind-driven processes in the evolution of warm marine cloud properties

Vishnu Nair[1], Edward Gryspeerdt[1], Antti Arola[2], Antti Lipponen[2], and Timo Virtanen[2]

[1]Department of Physics, Imperial College, London, UK
[2]Finnish Meteorological Institute, Kuopio, Finland

**Correspondence:** Vishnu Nair (v.nair16@imperial.ac.uk)

**Abstract.** The cloud droplet effective radius is a key variable when evaluating the interactions between aerosols and clouds. The activation of fine-sized sea salt from the ocean results in the formation of more but smaller cloud droplets (reducing the effective radius) in marine stratocumulus. Coarse sea spray aerosols are generated for high surface wind speeds and act as giant cloud condensation nuclei, which activate to form larger droplets. This increases the effective radius and initiates precipitation.

These high wind speeds also lead to enhanced moisture fluxes from the ocean surface. Although the opposing impacts of wind-driven fine and coarse marine sea spray aerosols have been documented, their observations have been limited to instantaneous satellite images. In this work, a novel framework is introduced that uses short-timescale observations of the temporal evolution of clouds to identify, isolate, and extract the process fingerprints of marine sea-salt and surface fluxes on stratocumulus cloud properties. This method shows that changes in droplet size previously attributed to aerosol are actually due to increases in

evaporation from high surface wind speeds. However, when this is accounted for, a clear impact of giant cloud condensation nuclei is observed, reducing cloud droplet number concentrations by initiating precipitation in polluted clouds. By isolating the causal aerosol impact on clouds from confounding factors, this method provides a pathway to improved constraints on the human forcing of the climate, whilst also demonstrating how marine aerosols limit the effectiveness of anthropogenic aerosol perturbations.

**1    Introduction**

Aerosols affect the Earth's radiation budget directly by reflecting and absorbing incoming solar radiation, and indirectly by acting as nucleation sites on which cloud droplets form (Twomey, 1974; Bellouin et al., 2020). Indirect effects occur by changing existing or new cloud properties and can have a cooling effect on global surface temperatures, hence offsetting a large part of the greenhouse gas warming (Stocker et al., 2013). This is by modifying the cloud reflectivity, both by affecting droplet

size and by driving time-dependent 'adjustments' (Albrecht, 1989), modifying the evolution of cloud properties (Glassmeier et al., 2021; Fons et al., 2023; Gryspeerdt et al., 2022). The effective radiative forcing from aerosol-cloud interactions (ACI) is the largest source of uncertainty in human forcing of the climate (Andreae et al., 2005; Bellouin et al., 2020). ACI contributions are mainly due to the instantaneous Twomey effect (Twomey, 1974) which affects the cloud microphysical properties, or due to adjustments to the cloud macrophysical properties.



The adjustments of low clouds, such as marine stratocumulus (MSC), to aerosol perturbations are crucial to the Earth's radiation budget (Slingo, 1990). There are significant changes in the budget for even a small change in MSC coverage and thickness, with even a 4% increase in the global area covered by low-level stratus clouds offsetting a 2-3K temperature increase from higher atmospheric $CO_2$ concentrations (Randall et al., 1984). Two key measures of the properties of clouds that affect its radiative properties and hence the effect of MSC on the ocean albedo are the liquid water path (LWP, a measure of the total

liquid water in a cloud), and the cloud droplet number concentration ($N_d$, a measure of the number of droplets in a cloud).

For a constant LWP, an increase in aerosol concentration (or cloud condensation nuclei, CCN) leads to an increase in $N_d$ and a decrease in the cloud droplet effective radius, $r_e$ (Twomey, 1977). More numerous smaller cloud droplets with a larger total droplet surface area reflect more sunlight, leading to an increase in cloud albedo. The decrease in $r_e$ can also modify the cloud macrostructure by suppressing precipitation (due to weakened collision-coalescence between droplets) which causes both the

LWP and albedo to increase (Albrecht, 1989). On the other hand, lower $r_e$ can also suppress in-cloud droplet sedimentation and enhance cloud-top radiative cooling which causes an increase in turbulent entrainment of free tropospheric air. Depending on the humidity of the entrained air, this can lead to a decrease or increase in LWP (Ackerman et al., 2004; Bretherto et al., 2007).

As well as the aerosol impact on cloud, cloud processes (such as precipitation) can modify aerosol. These feedback loops

of processes that occur simultaneously are difficult to unravel and are further dependent on different cloud and meteorological regimes, complicating the identification of causal aerosol impacts on cloud (Fons et al., 2023). There are differences in the estimates of the climate effects due to ACI from global climate models and observations. The accuracy of the representations of these separate adjustment processes in models is believed to be one of the reasons for this discrepancy (Mülmenstädt et al., 2024a, b). This creates a requirement for strong observational constraints on cloud processes, to ensure that models have

accurate representations of ACI.

There are multiple processes that can modify the cloud $r_e$, either by changing the cloud $N_d$ or via LWP. A key process via the $N_d$ pathway is the additional activation of cloud droplets on CCN from different sources: either entrained from the free troposphere, or aerosol produced by sea spray on the ocean surface (Wood et al., 2012). The role of sea spray aerosols is unique as the consequence of the ACI can vary depending on the size of sea salt generated. The cloud-top $r_e$ retrieved from satellite

observations has systematically higher values over the ocean than over land which has a higher fine anthropogenic aerosol (radii $< 1\mu$m) concentration (Bréon et al., 2002; Kaufman et al., 2005).

Both fine and coarse sea salt coexist, especially over the ocean. Perturbing clouds with fine sea salt (FSS) would lead to a reduction in $r_e$ thereby brightening clouds. However, coarse marine aerosols (CMA, radii $> 1\mu$m), which are generated for surface wind speeds greater than $4$ms$^{-1}$ (Lehahn et al., 2010), act as 'giant' CCNs and have been hypothesized to enhance

warm rain by accelerating the formation of larger cloud droplets (larger $r_e$) (Ludlam, 1951; Feingold et al., 1999; Lasher-Trapp et al., 2001; Rosenfeld et al., 2002; L'Ecuyer et al., 2009; Lehahn et al., 2011; Jung et al., 2015). By triggering rain and reducing the LWP, CMA can break up and hence reduce cloud reflectivity. However, other model studies have questioned this impact, showing that this depends on the aerosol concentration; CMA have a negligible impact on precipitation initiation in clean clouds (Dagan et al., 2015), or no impact at all (Blyth et al., 2013). More recent research (Liu et al., 2022; Liu et al.,



2025) suggests that there is an optimal combination which can effectively brighten clouds due to reduced $r_e$ and cloud cover. Although it is not an anthropogenic aerosol, sea-salt sets the background "unpolluted" state of the cloud, modifying the aerosol forcing. This makes it essential to correctly represent fine and coarse marine aerosol in any model used for future climate assessment.

The impact of different cloud adjustments to fine and coarse sea salt has potential implications for geoengineering through

marine cloud brightening (MCB). MCB deliberately injects clouds with aerosols (ideally seawater spray) to lower $r_e$ and increase reflectivity. In addition to setting the cloud background condition, thus determining how effective MCB can be in some conditions, the size distribution of the seeded aerosol now becomes important (Hoffmann and Feingold, 2021). Due to technical limitations, the seawater sprayed often contains coarse particles as well which can cause cloud breakup by initiating precipitation, making it important to consider the combined and opposing effects of fine and coarse sea salt and the possible

consequences in MCB projects.

Cloud $r_e$ modification occurs via changes in cloud LWP as well. An increase in cloud LWP leads to a a vertically deeper cloud which results in a higher cloud top $r_e$ under the adiabaticity assumption. At stronger surface wind speeds, there is more evaporation at the ocean surface and a consequent increase in surface moisture flux. This leads to a moist marine boundary layer, a lower cloud base, and the formation of thicker clouds with a larger LWP (Chen et al., 2011). Therefore, increased

low-level horizontal wind speeds can enhance the emission of fine sea salt and giant CCNs while at the same time evaporating and transporting more moisture into clouds. This makes the wind speed ($w_s$) a major driver of cloud change over the ocean through multiple pathways (figure 3). It is vital to distinguish between these two causal pathways ($w_s$ - $N_d$ - $r_e$ versus $w_s$ - LWP - $r_e$) and extract the "fingerprints" of these different processes to ensure accurate assessments of the climate response to anthropogenic aerosol changes.

Current observational studies are based on instantaneous satellite imagery. Recent observational assessments of the combined effects of FSS and CMA on cloud $r_e$ and warm rain (Liu et al., 2022), and cloud radiative effects (Liu et al., 2025) used instantaneous measures of the LWP, fixed to separate the combined effects of meteorological factors. However, studies including the evolution of nocturnal clouds over 12 hours (model studies, (Hoffmann et al., 2020; Glassmeier et al., 2021)) and short timescales (day time observations, (Gryspeerdt et al., 2022)) reveal that the LWP can evolve differently based on the

initial $N_d$ perturbation. The impacts of $N_d$ on LWP are not accurately captured by instantaneous measurements (Arola et al., 2022; Zhang et al., 2024). Temporal evolution of cloud properties provides a separate pathway to isolate aerosol impact on cloud (Meskhidze et al., 2009; Glassmeier et al., 2021; Gryspeerdt et al., 2021, 2022; Fons et al., 2023), removes any reliance on predetermined/instantaneous and possibly confounded (by meteorological and other cloud controlling factors) $N_d$ - LWP relationships, and ensures the accuracy of interpreted causal relationships (Zhang et al., 2024).

This work directly measures the impact of different processes on the evolution of $r_e$ over short time scales using a variety of observational data sets. A new framework is introduced to identify the role of different cloud processes in observations of cloud evolution. By assessing the impact of different meteorological properties on cloud evolution over a three hour time period, this framework isolates the individual fingerprints of fine and coarse sea salt, and surface fluxes, on LWP and $N_d$ changes. The analysis highlights the importance of the initial/background state of the cloud in the temporal evolution of its macrophysical



and microphysical properties. Non-aerosol processes have a more effective role in controlling the production of larger cloud droplets whereas coarse marine aerosols limit the overall impact of anthropogenic aerosols on cloud properties in very polluted conditions.

## 2 Data and Methods

### 2.1 Observational and meteorological reanalysis data

The properties of the cloud field are calculated from the Moderate Resolution Imaging Spectroradiometer (MODIS) cloud retrieval (Platnick et al., 2017) onboard both the Aqua and Terra satellites over the 10-year period 2007-17 (inclusive). The Terra and Aqua satellites provides information about cloud properties three hours apart from the two daytime overpasses. The cloud $N_d$ and LWP is derived from the values of $r_e$ and the cloud optical depth $\tau$ from the 2.1 $\mu$m retrievals (Quaas et al., 2006; Grosvenor et al., 2018), aggregated to a $1°\times1°$ resolution. Following (Gryspeerdt et al., 2021), boundary layer winds are used

to account for advection between the observation times of the different instruments, restricting analysis to gridboxes where $N_d$ and LWP are available for both instruments. The results in this work are from a region within the southeastern Atlantic stratocumulus deck between $40°$S to $10°$N and $30°$W to $10°$E.

Surface wind speeds are obtained from ERA5, The European Center for Medium-Range Weather Forecast Reanalysis v.5, which provides meteorological reanalysis products that offer an estimate of global atmospheric conditions that are collocated

with the MODIS observations (Hersbach et al., 2023).

Precipitating and non-precipitating cases are differentiated using the probability of precipitation (PoP) at each MODIS grid point, based on the proportion of liquid $r_e$ retrievals greater than $15\mu$m (Rosenfeld et al., 2002). This is calculated from the MODIS level 3 daily gridded product (MOD08_D3) using the $r_e$ histogram counts. The PoP is the ratio of small drops (sum of all histogram bins less than 15 $\mu$m) to all drops in the distribution/histogram (sum of all bins). If PoP > 70%, then the gridbox

is considered to be precipitating.

### 2.2 Difference of rates (DoR)

The temporal evolution of the $N_d$ and LWP ($dN_d/dt$ and $dLWP/dt$ respectively), is obtained by calculating the difference in properties between the two daytime satellite overpasses, which are approximately three hours apart- at 10:30 and 13:30 local solar time for Terra and Aqua respectively. The joint histograms for the relative rates of changes (in %/hr) in the $N_d$ - LWP

space are then generated by binning the cloud data according to their initial $N_d$ and LWP(Gryspeerdt et al., 2022). By strictly controlling the initial state of the cloud, we account for the confounding impact of existing meteorological variables which can change the properties of aerosol and the cloud simultaneously and could introduce spurious correlations on the development of Nd-LWP relationships.

Here, the difference-of-rates (DoR) method is introduced. The relative rates of change (in %/hr) are calculated by separating

or stratifying the cloud population based on a cloud/meteorological variable (such as surface (10 m) wind speeds, $w_s$, and



calculating the differences with respect to a reference data set. For example, in figure 2, the DoR (presented herewith with a symbol $\Delta$) with respect to $w_s$ is calculated as

$$\Delta_{w_s}\mathrm{d}N_d = \Delta_{(w_s^+)}\mathrm{d}N_d - \Delta_{(w_s^-)}\mathrm{d}N_d \tag{1}$$

where $w_s^+ = w_s > 4$ ms$^{-1}$ and $w_s^- = w_s < 4$ ms$^{-1}$. Here, the cloud population with $w_s^-$ is the reference data set. Multiple
DoRs can be calculated by splitting $w_s^+$ into smaller bins/data ranges (different columns in figure 2).

## 3    Results and Discussion

### 3.1    The precipitation fingerprint

In addition to acting as a sink for the cloud $N_d$ through the sedimentation of droplets, precipitation plays a key role in the scavenging of CCN, which in turn can reduce $N_d$. The effects of precipitation are seen primarily in the (upper) left quadrant
(LWP $> 50$gm$^{-2}$, $N_d < 50$cm$^{-3}$), i.e., for clouds with a high initial LWP and a low $N_d$. The DoRs between precipitating and non-precipitating clouds (figure 1c) show that precipitation acts as a sink for the cloud $N_d$, with a reduction of $N_d$ observed in more strongly precipitating cases. This leads to a smaller overall net increase in $\mathrm{d}N_d$ (i.e., the change in $N_d$ over three hours) for precipitating clouds (lighter reds and darker greens in figure 1a), and a smaller decrease in $\mathrm{d}N_d$ for non-precipitating clouds (darker reds and lighter greens, in figure 1b). Consequently, the corresponding DoR, $\Delta_{PoP}\mathrm{d}N_d$, is negative (figure 1c).

The few positive values (red) in the DoR field in figure 1c are possibly non-precipitating clouds which were not filtered out using the $r_e$ threshold. The positive and negative regions in figures 1a and b may be partly driven by the regression to the mean effect by using a filter in $r_e$, which is also used to calculate $N_d$ and LWP. Similar patterns were obtained as in the figures 1 a-c when an independent data source (Eastman et al., 2019) for precipitation was used (supplementary section). The MODIS filters were subsequently chosen to identify precipitating clouds for the rest of the analysis. Using the effective radius as a measure
of precipitation allows the precipitation data at the start of the time step to be included, such that the impact of precipitation on the cloud evolution is identified (rather than the impact of cloud evolution on precipitation, as is obtained using precipitation from the later overpass at the end of the timestep).

In non-precipitating clouds, the $r_e$ is smaller, and there are more, smaller cloud droplets in the interfacial layer at the cloud top. This means that the entrainment of free tropospheric air is more effective in evaporating droplets at the cloud top, leading
to a higher decrease (or a smaller increase) in dLWP in these clouds (Bretherto et al., 2007). Consequently, the DoR for LWP is positive in the non-precipitating clouds in figure 2f. In contrast, strongly precipitating clouds lose more liquid water resulting in a more negative (less positive) change in LWP. This results in a negative $\Delta_{PoP}$dLWP for these clouds in figure 1f, corresponding to the strongly negative region in figure 1c for $N_d$. There is also a scattered positive presence over this negative region. These suggest cases where non-precipitating clouds can have a more negative (less positive) dLWP compared
to precipitating clouds.





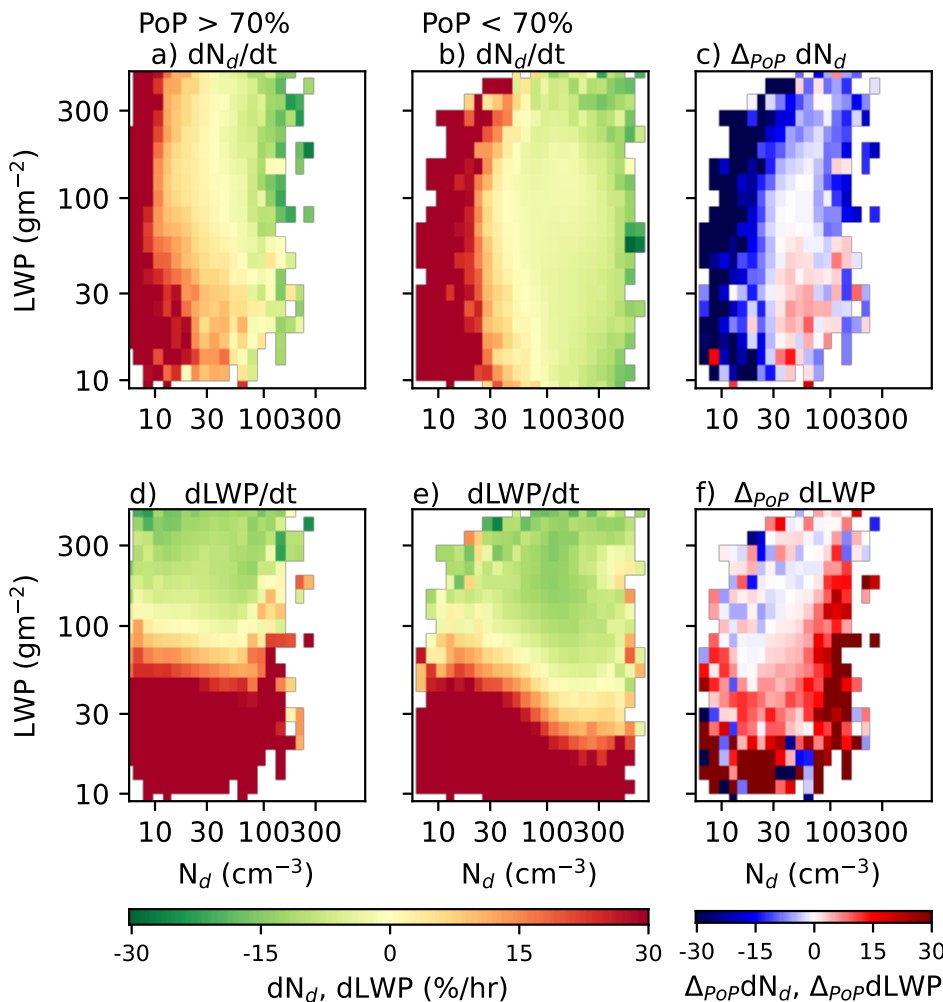

**Figure 1.** The effect of precipitation on d$N_d$/d$t$ (a,b) and dLWP/d$t$ (d,e). (a) d$N_d$/d$t$ for precipitating clouds(PoP > 70%), (b) d$N_d$/d$t$ for non-precipitating clouds (PoP<70%), (c)$\Delta_{PoP}$d$N_d$, the difference between sub figures a and b, (d) dLWP/d$t$ for precipitating clouds (PoP > 70%), (d) dLWP/d$t$ for non-precipitating clouds (PoP<70%), (f)$\Delta_{PoP}$LWP = d-e.



**Figure 2.** Effect of surface wind speed on $N_d$ and LWP. All figures show DoRs, $\Delta dN_d = \Delta_{(w_s^+)}dN_d - \Delta_{(w_s^-)}dN_d$, where $w_s^- = w_s < 4\text{ms}^{-1}$ and a) $w_s^+ = 4 < w_s < 7\text{ms}^{-1}$, b) $w_s^+ = 7 < w_s < 9\text{ms}^{-1}$, c) $w_s^+ = 9 < w_s < 12\text{ms}^{-1}$, d) $w_s^+ = w_s > 12\text{ms}^{-1}$ for precipitating clouds. Figures (e) - (h) are $\Delta dN_d$ for non-precipitating clouds, and figures (i) - (l) are $\Delta$dLWP for non-precipitating clouds for similar $w_s$ ranges.





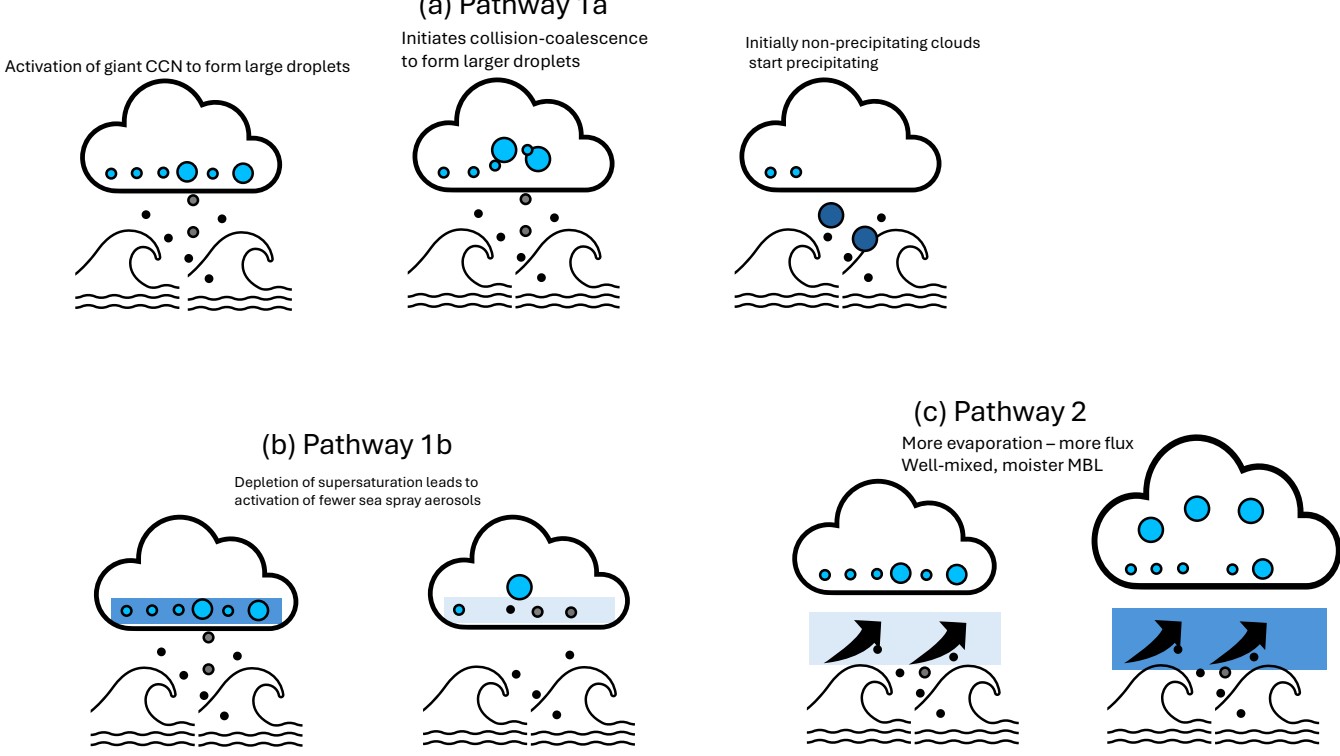

**Figure 3.** Different pathways to explain the $w_e$ - $N_d$/LWP - $r_e$ correlation.

## 3.2 Disentangling the impact of wind driven processes on cloud effective radius

With the new framework effectively extracting the precipitation fingerprint, we apply this technique to identify the different processes that modify the $r_e$. To investigate the simultaneous effects of FSS and CMA in altering $r_e$ through changes in $N_d$, the DoRs are calculated by stratifying the data by surface wind speeds. Horizontal low-level (10m) winds have been shown

to be strongly correlated with sea salt production (Lewis and Schwartz, 2004). In contrast to figure 1, all the plots in figure 2 are DoRs, $\Delta_{w_s}\mathrm{d}N_d$ and $\Delta_{w_s}\mathrm{d}$LWP, calculated for different wind speed ranges and with respect to a reference dataset with $w_s < 4\mathrm{ms}^{-1}$ (as detailed in the methods section). Additionally, we treat initially precipitating and non-precipitating clouds separately as the impact of aerosol-induced precipitation suppression works differently for these two categories.

For initially precipitating clouds, precipitation suppression is expected to increase at higher $w_s$ with the introduction of

additional FSS. Therefore, $\mathrm{d}N_d$ would tend to be less (more) negative (positive) as $w_s$ increases (due to reduced in-cloud scavenging). Consequently, $\Delta_{w_s}\mathrm{d}N_d$ would be negative (blue) as seen in the left quadrant in figures 2a to d.

A positive fingerprint, consistent with FSS acting as CCN, strengthens with wind speed and dominates the precipitation effect at wind speeds greater than 12 ms$^{-1}$ in figure 2d. In addition to the increased aerosol burden, high $w_s$ leads to stronger updrafts at the cloud base (Colón-Robles et al., 2006) increasing the activation of FSS and the formation of new droplets.





The effect of CMA is seen in non-precipitating clouds with a high initial $N_d$ ($> 100 \text{cm}^{-3}$) in figures 2e - h, with a negative trend in $\Delta_{w_s}\text{d}N_d$ consistent with the results from (Feingold et al., 1999; Yin et al., 2000). The CMA are usually the first to activate and form droplets at the cloud base. This depletes the supersaturation here, thereby inhibiting the activation of FSS into smaller droplets. This skews the droplet size distribution to larger sizes, enhancing the collision-coalescence rate (Rosenfeld et al., 2002). Both processes lead to a higher $r_e$ which is a key precursor to precipitation initiation.

A negative $\Delta_{w_s}\text{d}N_d$ region is consistent with a giant CCN-induced reduction in $\text{d}N_d$. In addition to fewer new activated droplets, the smaller $\text{d}N_d$ could also be due to giant CCN-induced precipitation (in initially non-precipitating clouds). Once giant CCNs are activated, condensational growth and collision-coalescence to raindrop sizes are expedited within this time scale (of three hours). Increasing $w_s$ leads to the formation of more CMA which shows up as a stronger signal in the DoR. The two effects are illustrated in figure 3a and b. As expected, there is no perceptible impact of CMA on already raining clouds as

drizzle is already active.

A relationship between CMA and $r_e$ by altering $N_d$ can clearly be seen. However, there is a strong positive signal in $\Delta\text{dLWP}$ across all initial $N_d$ suggesting an alternate pathway to explain the $w_s$ - $r_e$ correlation. Stronger surface winds over the ocean lead to an increase in surface fluxes through increased evaporation (Kazil et al., 2016). This moistens and deepens the marine boundary layer. In addition, higher wind speeds enhance wave formation - these waves can break and produce white caps and

sea spray, which enhances the sea-to-air latent heat flux. This is an additional source to the cloud LWP leading to thicker clouds with larger droplets, and corresponding larger dLWP over the observed period (figures 2i - l).

## 4    Conclusions and Outlook

This article highlights the effectiveness of two different pathways through which surface wind-driven processes can modify the cloud droplet effective radius $r_e$. Using observations of the temporal evolution of clouds, process fingerprints of the effects of

marine aerosols on the cloud properties were extracted. Giant CCNs were shown to reduce cloud $N_d$ (figures 2 e -h) either by

–   depleting supersaturation by activating to form larger droplets. This increases competition for supersaturation among the remaining CCNs leading to the activation of fewer new droplets.

–   initiating precipitation in clouds with an initially high droplet number concentration.

Contrary to previous results, in addition to fine and giant CCNs, we identify and highlight the role of wind-driven surface

fluxes in the thickening of marine stratocumulus clouds and therefore in the increase of $r_e$. A change of -2.5% in $\text{d}(\ln N_d)$ (for $N_d > 100$ in figure 2f) due to pathway 1 (figures 3a,b), and 5% in $\text{d}(\ln\text{LWP})$ (for $N_d > 100$ in figure 2j) due to pathway 2 (figure 3c) translates into an increase in $\text{d}(\ln r_e)$ of 7.5% and 30% respectively (details in the appendix). This clearly identifies the second pathway - via increased surface fluxes - as the more dominant physical process that increases $r_e$, while simultaneously obscuring the impact of marine aerosol on clouds.

The new framework introduced in this work addresses this issue by extracting the individual process fingerprints of both fine and coarse marine aerosols. While retrieval biases can still affect the quantification of the initial state, the focus on time



development reduces the impact of correlated errors in $N_d$ and LWP that affect previous studies (Arola et al., 2022). Further studies are required to assess the impact of other factors on these fingerprints, particularly the diurnal cycle (Zhang et al., 2024), but this method has already been effective at isolating the varying effects of different aerosol types on cloud evolution. The

impact of fine sea salt follows previous observational studies, increasing $N_d$, but coarse marine aerosol is shown to decrease $N_d$, particularly in initially non-precipitating cases with a high $N_d$. This shows that even small amounts of coarse aerosol can limit the efficacy of anthropogenic aerosol injections, providing an important constraint on the cloud response to aerosol and limiting the effectiveness of proposed marine cloud brightening programs.





**Appendix A: Investigating regression to the mean effects using an independent data source for precipitation**

It is possible that the positive and negative regions in figures 1a and b are partly driven by the regression to the mean effect. Whereby, a positively biased first measurement would likely be followed by a smaller second measurement. By applying a threshold on $r_e$ (which is used to calculate both $N_d$ and LWP) to distinguish between precipitating and non-precipitating clouds, there is a chance that similar regression to mean effects are inadvertently introduced in the DoRs in figure 1c.

We perform an alternate analysis using an independent data source for precipitation: warm rain rates inferred from AMSR/E
and AMSR/2 89GHz passive microwave brightness temperatures trained using CloudSat rain rate observations (Eastman et al., 2019). The results for DoRs from the dataset provided by Eastman et al. (2019) suggest patterns similar to those in figures 1 and 2. This suggests that the patterns are indicative of precipitation effects rather than the regression to the mean (which might still be at play but less dominant). We refrain from using the data set from Eastman et al. (2019) to identify precipitating clouds in the main manuscript as these are collocated with data from Aqua, which is at the end of the time step in the context
of this manuscript. By using the effective radius as a measure of precipitation we are using more information of the cloud microphysics and obtain data from the start of the time step. This allows us to identify the role of precipitation and other processes during the evolution of the cloud. Original CloudSat rain rate observations were also considered, but these are too sparse/patchy to provide reliable results.

**Appendix B: Cloud $r_e$ sensitivities to changes in LWP and $N_d$**

The cloud $N_d$ and LWP are calculated using

$$N_d = \frac{\sqrt{5}}{2\pi k \sqrt{\rho_l Q}} \sqrt{f_{ad}\Gamma} \tau^{1/2} r_e^{-5/2},$$

$\text{LWP} = 5/9 r_e \rho_l \tau.$

The change in $r_e$ with a change in $N_d$ can be represented as

$$\frac{\partial \ln r_e}{\partial \ln N_d}\Big|_{\text{LWP}} = -\frac{1}{3},$$
$$\frac{\partial \ln r_e}{\partial \ln \text{LWP}}\Big|_{N_d} = \frac{1}{6}.$$



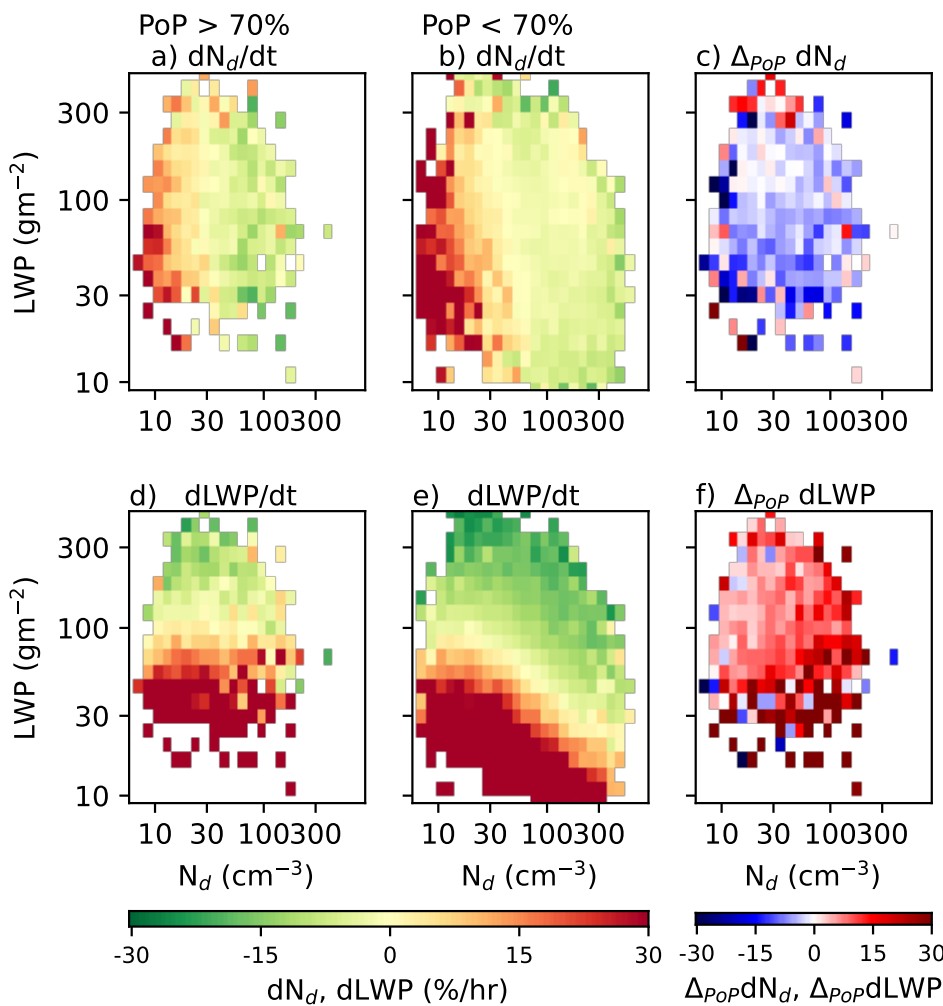

**Figure A1.** Calculation of DoRs as in figure 1 using CloudSat data.





**Figure A2.** Effect of surface wind speed on $N_d$ and LWP. As in figure 2 but with CloudSat data.





*Data availability.* The data supporting the conclusions, along with the code used for processing this data and generating the figures in this study are available with the identifier https://doi.org/10.5281/zenodo.16882487.

*Author contributions.* EG and VN designed the study. VN performed the analysis and wrote the paper. All of the authors assisted in the interpretation of the results and commented on the paper.

*Competing interests.* The authors declare that no conflicting interests are present.

*Acknowledgements.* The authors acknowledge funding from Horizon Europe programme under Grant Agreement No 101137680 via project CERTAINTY (Cloud-aERosol inTeractions & their impActs IN The earth sYstem).



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
