# Peer review of "Observing the role of wind-driven processes in the evolution of warm marine cloud properties"

_EGUsphere, 2025_

## Referee Comment (RC1)

**Review of "Observing the role of wind-driven processes in the evolution of warm marine cloud properties" by Nair et al. (egusphere-2025-4272)**

This study analyzes the impact of surface winds on aerosol-cloud-interactions, focusing on the production of sea salt aerosols and the surface latent heat flux. For this, the authors used consecutive observations of the Aqua and Terra satellite that enabled them to determine the temporal change in the droplet number concentration ( $N_d$ ) and liquid water path (LWP). While I generally like the approach and tend to agree with the results presented here, the study is too superficial to be considered for publication. Thus, I have to recommend rejection for now, but like to encourage the authors to resubmit a revised manuscript.

**Major Comments**

Diurnal cycle. While I believe that the methodology might limit the influence of diurnal cycle on changes in  $N_d$  and LWP, I am wondering if it affects constraining the data by wind speed if there are distinct changes in wind speed due to the diurnal cycle between the Aqua and Terra overpasses?

Are sea salt aerosols the only relevant CCN source? While I agree that sea salt aerosols are an important aerosol source, increased entrainment might make the free troposphere as a stronger CCN source. Moreover, the ocean could also be source for sulfate aerosols. Chemical processing in the boundary layer could also increase the size of smaller aerosols, turning them into potential CCN.

Effects on boundary layer dynamics. While it has been touched upon at several places, a stronger focus on the effects of surface wind speed on boundary layer dynamics might be necessary. Increasing surface wind speed will also increase shear, which increases turbulence and hence entrainment rates, which could reduce the LWP.

Description of Methodology. The methodology (Sec. 2) needs some attention. While I understand the concept, I do not understand the details. How are relative changes determined? Based on I. 117, the rates of  $N_d$  and LWP should have units of  $1/cm^3/s$  and  $g/m^2/s$ , respectively. How is the DoR independent of time? Are the aforementioned rates multiplied with a timescale? What is the timescale? In Appendix B, several quantities need to be explained (e.g., Q).

**Minor Comments**

- Ll. 2 4: How are fine sea salt aerosols created? Also by wind?
- L. 10: Evaporation of what?
- L. 32: Define effective radius.
- L. 36: How does a smaller effective radius enhance cloud top cooling? As long as the LWP is sufficiently high (>  $30 \text{ g/m}^2$ ), the emission of longwave radiative cooling from warm clouds does not depend on microphysical details.
- LI. 59 60: I believe that only an increase in cloud cover can increase the scene albedo. A reduction in cloud cover usually results in a darkening if it is not accompanied by a massive increase in cloud albedo.
- Ll. 90 91: "directly measures" is a bit of an overstatement.
- Ll. 111 115: Does this indicate that a MODIS grid point consists of several measurements of  $r_e$ ? In other words, is the MODIS grid point an average?
- Ll. 141 143: I do not understand the impact of "regression to the mean", and how it is related to the calculation of  $r_e$  and  $N_d$ . Please clarify.
- Ll. 148 150: This is not how the sedimentation-entrainment feedback explained by Bretherton et al. (2009) works.
- L. 150: Should this be a decrease in LWP, not dLWP?

Ll. 150 – 151: The DoR is calculated from the difference not-precipitating and precipitating clouds. Referencing to non-precipitating clouds feels strange.

L. 166: I would have expected  $\Delta dN_d$  to become increasingly positive (red), as seen in Fig. 2d. Why is it negative (blue) for Figs. 2a to c?

Ll. 175 – 180: Please also show  $\Delta dLWP$  for precipitating clouds.

L. 181 – 186: How does the LWP increase? Due to a an increase in cloud top height by stronger entrainment, or due to a decrease in cloud base height due to more condensation?

**Technical Comments**

Ll. 37 ff.: "Bretherton", not "Bretherto".

Ll. 104 ff.: Citation style is wrong.

L. 144: "effective radius" to "re".

L. 151: Reference to Fig. 1f, not 2f?

Figs. 2 and A2: Change labels  $dN_d$  to  $\Delta dN_d$  and dLWP to  $\Delta dLWP$ . Are the DoR units correct?

Appendix A: Please refer explicitly to Figs. A1 and A2.

**References**

Bretherton, C.S., Blossey, P.N. and Uchida, J., 2007. Cloud droplet sedimentation, entrainment efficiency, and subtropical stratocumulus albedo. *Geophysical Research Letters*, 34(3).

---

## Author Comment (AC1)

**Referee 1:**

Major Comments

Diurnal cycle. While I believe that the methodology might limit the influence of diurnal cycle on changes in Nd and LWP, I am wondering if it affects constraining the data by wind speed if there are distinct changes in wind speed due to the diurnal cycle between the Aqua and Terra overpasses?

*By constraining data by wind speed (in addition to initial Nd and LWP), we are bringing together similar clouds evolving under similar meteorological conditions at which aerosol perturbations occur. This allows us to isolate the role of surface wind speeds (which is in turn related to sea salt size and concentration) on cloud Nd and LWP evolution. The dLWP and dNd are for a given* **initial** *Nd, LWP and wind speed, and any change in the surface wind speeds between terra and aqua due to the diurnal cycle does not impact the interpretation of this analysis which focuses on the impact of variations in the initial conditions.*

*Simultaneously, driven by this reviewer's comment, we have also looked at the changes in $w_s$ between Aqua and Terra observations for a year (2007), and plotted a joint histogram for $dw_s$ ( the change in wind speed between aqua and terra) below. The changes are indeed very small (<0.1 m/s) and spread uniformly over the Nd-LWP phase space.*

[Figure]

Are sea salt aerosols the only relevant CCN source? While I agree that sea salt aerosols are an important aerosol source, increased entrainment might make the free troposphere as a stronger CCN source. Moreover, the ocean could also be source for sulfate aerosols. Chemical processing in the boundary layer could also increase the size of smaller aerosols, turning them into potential CCN.

*We agree that multiple sources for aerosols exist over the ocean. However, we expect sea salt to be the one that is most correlated with surface wind speeds and hence drive the conclusions using the DoR method.*

*Free troposphere as a source – As explored in detail in the next comment on wind shear and cloud-top entrainment, we do not expect a clear increased cloud top entrainment due to increased surface wind speeds. Therefore, even though the free troposphere is a source of CCN,*

*it will not drive variations in cloud evolution as a function of wind speed and can be discounted from our analysis.*

*Sulfate aerosols – A detailed discussion has been added in the conclusion (L301). A mention has also been made of the different types of aerosols in the introduction (L49).*

Effects on boundary layer dynamics. While it has been touched upon at several places, a stronger focus on the effects of surface wind speed on boundary layer dynamics might be necessary.

Increasing surface wind speed will also increase shear, which increases turbulence and hence entrainment rates, which could reduce the LWP.

*We agree that wind shear enhances entrainment thereby thickening the entrainment zone (the region of negative buoyancy flux at the cloud top) and deepening the stratocumulus topped boundary layer. However, the dominant contributor to this will be the shear-generated turbulence in the entrainment zone, rather than the shear-generated turbulence in the surface layer which is then transported to the entrainment zone. We expect the surface wind speeds used in this study to directly increase shear generated at the surface; there is varying evidence of this increasing entrainment at the cloud top. We have added a detailed discussion in the conclusions (L315):*

*'It is important to acknowledge the possible role of surface winds on stratocumulus topped boundary layer (STBL) dynamics. Specifically, the role of surface shear on cloud-top entrainment rate. Entrainment of free tropospheric (FT) aerosols across the cloud-top entrainment interfacial layer (EIL) can lead to the introduction of CCN. This possibly affects our results if a direct correlation exists between wind-generated surface shear and entrainment rates. However, even though there is considerable evidence of wind shear across the **inversion** enhancing the entrainment rate of dry, warm FT air and reducing the cloud fraction and LWP (Schulz2018, Wang2008, Wang2012, Zapata2021), the effect of surface shear on entrainment in STBL is still unclear. Increased low level vertical wind shear can contribute to a turbulent and well-mixed STBL. However, any increase to entrainment (a turbulent kinetic energy (TKE) consuming process) is dependent on the availability of the surface shear-generated TKE at the EIL. This surface shear generated TKE must be transferred through the entire depth of the STBL to the EIL as it goes through the energy cascade process of turbulence, before it can be used to drive entrainment. However, studies on the STBL specifically looking at the role of surface shear and cloud top shear failed to see conclusive evidence on an increase in entrainment rates due to **only** surface shear (Zapata2021).*

*Studies on the interaction of a constant large-scale wind speed with the STBL showed that it is possible through buoyancy driven dynamics (rather than shear driven) for geostrophic wind to promote STBL growth and enhance entrainment throughout the diurnal cycle (Kazil2016). Higher surface moisture flux at increased wind speed boosts the latent heat release and buoyant production of TKE in cloud updrafts leading to increased entrainment. At the same time, they point out that features of boundary layer dynamics that determine entrainment exist, but require more in-depth study.*

*This suggests that we cannot completely discount the possibility that wind driven entrainment of aerosols from the FT can affect the results in this work, especially in figure 2.*

*Additionally, depending on the humidity of the entrained FT air, there could be an increase or decrease in cloud LWP (Ackerman2004). Accounting for the contribution of surface winds to cloud-top entrainment and hence Nd and LWP would require controlling for both FT aerosol concentration and relative humidity (in addition to wind speeds) which complicates the analysis using DoRs. Since we expect the ws - surface shear - entrainment correlation to be weak, and the ws - sea salt correlation to be strong and dominate the Nd - LWP phase space, we reserve this analysis for future work.'*

Description of Methodology. The methodology (Sec. 2) needs some attention. While I understand the concept, I do not understand the details. How are relative changes determined? Based on l. 117, the rates of Nd and LWP should have units of 1/cm3/s and g/m2/s, respectively. How is the DoR independent of time? Are the aforementioned rates multiplied with a timescale? What is the timescale? In Appendix B, several quantities need to be explained (e.g., Q).

*The methods section has been updated to address this comment (L111). All terms in the appendix have also been defined. The relative changes are calculated as below:*

*The temporal evolution of the Nd and LWP (dNd/dt and dLWP/dt respectively), is obtained by calculating the difference in properties between the two daytime satellite overpasses, which are approximately three hours apart- at 10:30 and 13:30 local solar time for Terra and Aqua respectively. %for the relative rates of changes (in %/hr)*

*Joint histograms in the Nd - LWP space are then generated by binning the cloud data as a function of their initial Nd and LWP {Gryspeerdt-2022}. These are then converted to relative rates of changes by dividing the differences first with the bin widths (Nd bins for dNd, and the LWP bins for dLWP), and then with the time step of 3 hours (time difference between Aqua and Terra observations). The final result is expressed as a percentage with units of (\%/hr) by multiplying with 100.*

Minor Comments

Ll. 2 – 4: How are fine sea salt aerosols created? Also, by wind?

*Yes. This has been clarified in L58.*

L. 10: Evaporation of what?

*Corrected to 'evaporation from the ocean surface due to high surface wind speeds' (L10)*

L. 32: Define effective radius.

*The cloud droplet effective radius, re, is physically the area weighted mean of the cloud droplet size distribution. Added to line L31.*

L. 36: How does a smaller effective radius enhance cloud top cooling? As long as the LWP is sufficiently high (> 30 g/m2), the emission of longwave radiative cooling from warm clouds does not depend on microphysical details.

*Corrected to : 'enhanced evaporative cooling'. Line L37.*

Ll. 59 – 60: I believe that only an increase in cloud cover can increase the scene albedo. A reduction in cloud cover usually results in a darkening if it is not accompanied by a massive increase in cloud albedo.

*This was a typo. Should be 'due to reduced re and enhanced cloud cover/fraction'. L68.*

Ll. 90 – 91: "directly measures" is a bit of an overstatement.

*Deleted 'directly' - L97.*

Ll. 111 – 115: Does this indicate that a MODIS grid point consists of several measurements of re? In other words, is the MODIS grid point an average?

*Yes, these are L3 products, so data at a pixel level aggregated to a 1° x 1° latitude-longitude grid. The methodology has been expanded to reflect this. (L115)*

Ll. 141 – 143: I do not understand the impact of "regression to the mean", and how it is related to the calculation of re and Nd. Please clarify.

*A positively biased first measurement (of re) is followed by a smaller second measurement which is the regression to the mean effect. Since re is positively correlated with LWP and negatively correlated with Nd, this shows up as a highly negative dLWP and a large positive dNd. And an opposite effect for an initially negatively biased measurement of re. Now, by using the same re values as a filter to separate precipitating and non-precipitating clouds (rather than an independent source of measure of precipitation), we might be inadvertently introducing regression to the mean effects in the DoRs also.*

*The following more general and detailed description of the regression to the mean effect is added to the Methods section (L142):*

*'If the clouds are advected across regions with a large gradient in meteorological properties, this would result in a large change in the cloud properties owing to how correlated the cloud is to a strong climatological change. Clouds with a high (low) initial value of LWP or Nd is likely to show a decrease (increase) in LWP or Nd, which is consistent with a 'regression to the mean' effect. This can happen as a statistical effect, where even when the cloud is remaining stationary, a positively biased first measurement (of re) is followed by a smaller second measurement. Since re is positively correlated with LWP and negatively correlated with Nd, this shows up as a highly negative dLWP and a large positive dNd. And an opposite effect for an initially negatively biased measurement of re. Previous studies (Eastman et al 2016a, Eastman et al 2016b) have successfully*

*accounted for this by looking at anomalous changes across Lagrangian trajectories by removing seasonal means for day and night separately. On the other hand, it was shown in a previous study (Gryspeerdt et al 2021) (where dLWP and dNd are calculated from MODIS Terra and Aqua) that the 'flowfields' (the rate of change of Nd and LWP) do not look the same when dLWP and dNd are binned by the final LWP and Nd. If this was indeed a regression to the mean effect, the flowfields should have looked the same when calculated from either direction. As stated in Gryspeerdt et al 2021, while this does not completely rule out the impact of retrieval biases and the regression to the mean effect, it does rule out the possibility of the results being a statistical artefact caused by random biases.'*

Ll. 148 – 150: This is not how the sedimentation-entrainment feedback explained by Bretherton et al. (2009) works.

*This has been rewritten as follows: L187*

*Droplet sedimentation at the cloud-top interfacial layer (EIL) depletes liquid water from this zone, leading to reduced entrainment and thicker clouds (high LWP) (Ackerman2004, Bretherton2007). In precipitating clouds the $r_e$ is generally higher leading to droplets sedimenting out of the EIL. Conversely, in not-precipitating clouds, the $r_e$ is smaller, and there are more, smaller cloud droplets in the interfacial layer at the cloud top. This leads to evaporative enhancement of entrainment of free tropospheric air, leading to a thinner cloud layer or a higher decrease (or a smaller increase) in dLWP in not-precipitating clouds compared to precipitating ones.*

*Consequently, the DoR for LWP is positive in figure 2 everywhere except in the region corresponding to the strongly negative region in figure 1c for Nd.*

*Here, strongly precipitating clouds lose more liquid water (as rain) resulting in a more negative (less positive) change in LWP.*

L. 150: Should this be a decrease in LWP, not dLWP?

*Corrected to 'decrease in LWP'. L194*

Ll. 150 – 151: The DoR is calculated from the difference not-precipitating and precipitating clouds. Referencing to non-precipitating clouds feels strange.

*Changed to 'not-precipitating' throughout the manuscript.*

L. 166: I would have expected ΔdNd to become increasingly positive (red), as seen in Fig. 2d. Why is it negative (blue) for Figs. 2a to c?

*Precipitation suppression would result in a less negative dNd as ws increases. Since ΔdNd is the difference with the ws<4 dataset (where the effect of suppression would be the least – hence a very negative dNd), the dNd of figure 2a (and also b,c) will be more negative than that of dNd(ws<4). This would mean the ΔdNd( a less negative dNd – a*

*very negative dNd (of ws<4 dataset)) should still be negative/blue in figures a-d. In fig 2d, the effect of activation of new droplets from FSS acting as CCN possibly dominates this effect leading to a positive dNd and ultimately a positive ΔdNd.*

Ll. 175 – 180: Please also show ΔdLWP for precipitating clouds.

*Added to figure 2.*

L. 181 – 186: How does the LWP increase? Due to an increase in cloud top height by stronger entrainment, or due to a decrease in cloud base height due to more condensation?

*The LWP increase is likely due to a combination of the above two reasons. However, the current methodology does not allow us to answer this question clearly as this has been mentioned in the manuscript. (L230)*

*'This can increase the cloud depth either by lowering the cloud base through condensation or increasing the cloud top height by entrainment (driven by buoyant production of kinetic energy in the updrafts {Kazil2016}). The current methodology does not allow us to distinguish between these two effects.'*

Technical Comments

Ll. 37 ff.: "Bretherton", not "Bretherto".

*Corrected.*

Ll. 104 ff.: Citation style is wrong.

*Corrected.*

L. 144: "effective radius" to "re".

*Corrected.*

L. 151: Reference to Fig. 1f, not 2f?

*Corrected.*

Figs. 2 and A2: Change labels dNd to ΔdNd and dLWP to ΔdLWP. Are the DoR units correct?

*Changed.*

Appendix A: Please refer explicitly to Figs. A1 and A2.

*Corrected.*

References

Bretherton, C.S., Blossey, P.N. and Uchida, J., 2007. Cloud droplet sedimentation, entrainment efficiency, and subtropical stratocumulus albedo. Geophysical Research Letters, 34(3).

*Corrected.*

---

## Author Comment (AC2)

**Referee 2**

Main points:

-Can you go into a little more detail concerning how you link the Aqua and Terra observations? Are you using trajectories to link each 1x1 degree box in Terra to their Aqua counterpart? What level winds are you using, and what assumptions are you making about these links?

*The cloud field is advected (from Terra to Aqua, 10.30 to 13.30) using ERA5 reanalysis wind fields at 1000hPa. The advection is calculated on a $0.25° \times 0.25°$ resolution grid as the movement of the cloud fields over 3h is expected to be less than $1°$. Each grid box on the fine grid is treated as a 'parcel trajectory' and advected using the wind fields. The Aqua data are then sampled at the end points of these trajectories and then aggregated to a $1° \times 1°$ grid.*

*The ERA5 wind field at 1000hPa is used as this level has been shown to accurately predict the locations of ship tracks, and hence suitable for advecting clouds over short time scales.*

*The above details have been added to the manuscript (L119).*

-Data section: Are you doing any sort of filtration based on MODIS cloud cover or ERA5 meteorology? It is possible **that differing cloud morphologies** may have differing dominant processes, or that Nd and LWP anomalies could be associated with differing meteorology. It would be comforting to test for this, to see if these relationships persist when controlling for variables like cloud cover, EIS, SST, and humidity above the cloud.

*We completely agree that the effect of cloud morphologies and large-scale meteorology can have confounding influences on the Nd-LWP relationship. Here, we are looking at simpler cases, and the effects of other variables such as EIS, SST and free tropospheric humidity are reserved for future studies. These can affect cloud top entrainment which could indirectly affect the results by introducing new FT aerosols. We have included a discussion of this in the conclusions now (L324). We decided to focus on the role of wind-driven processes as this will have a more prominent role in answering the question of the interplay between fine and coarse marine aerosols.*

-LWP and Nd have strong variability with the MODIS sensor view angle. Have you looked at these relationships while controlling for this view angle? I recommend sub-setting the data into high/low sensor zenith angle bins and checking for consistency, as the zenith angle biases can be extremely strong.

*We have already accounted for the MODIS sensor view angle by only considering pixels with sensor angles $> 55°$ and solar zenith angle $> 65°$ following Grosevnor et al 2018. The data and methods section have been updated to reflect this and other details on filtering applied to MODIS data (L113).*

-Line 141 concerning regression to the mean: The patterns of increase/decrease seen in Figures 1a-b and 1d-e are almost certainly driven by regression to the mean, as stated. Anomalous initial values along trajectories have a strong tendency to regress to the mean as shown here:

Eastman, R., Wood, R., Bretherton, C.S., 2016. Time Scales of Clouds and Cloud-Controlling Variables in Subtropical Stratocumulus from a Lagrangian Perspective. https://doi.org/10.1175/JAS-D-16-0050.1

This figure shows this behavior clearly, with Low Nd and Low LWP adjusting positively as you move forward in time, and vice-versa for high values. That paper and others from that group explore how to deal with these tendencies, using a similar technique to the DoRs used here. It may be good to mention those papers to show how a similar technique has been successful in the past.

*The following paragraph has been added to the methods section while introducing the DoR technique (L142):*

*If the clouds are advected across regions with a large gradient in meteorological properties, this would result in a large change in the cloud properties owing to how correlated the cloud is to a strong climatological change. Clouds with a high (low) initial value of LWP or Nd is likely to show a decrease (increase) in LWP or Nd, which is consistent with a 'regression to the mean' effect. This can happen as a statistical effect, where even when the cloud is remaining stationary, a positively biased first measurement (of re) is followed by a smaller second measurement. Since re is positively correlated with LWP and negatively correlated with Nd, this shows up as a highly negative dLWP and a large positive dNd. And an opposite effect for an initially negatively biased measurement of re. Previous studies (Eastman2016a, Eastman:2016b) have successfully accounted for this by looking at anomalous changes across Lagrangian trajectories by removing seasonal means for day and night separately. On the other hand, it was shown in a previous study (Gryspeerdt et al 2021) using dLWP and dNd calculated from MODIS Terra and Aqua that the 'flowfields' (the rate of change of Nd and LWP) do not look the same when dLWP and dNd are binned by the final LWP and Nd. If this was indeed a regression to the mean effect, the flowfields should have looked the same when calculated from either direction. As stated in Gryspeerdt et al 2021, while this does not completely rule out the impact of retrieval biases and the regression to the mean effect, it does rule out the possibility of the results being a statistical artefact caused by random biases.*

-In figure 2e-l there appears to be lots of noise on the left sides of the distributions. Is this signal believable? Larger bins or some sort of significance test may eliminate some of this distracting noise.

*We agree that the region with Nd<30 (approximately) is very noisy. This is why we have restricted any conclusions from this figure for Nd>50. We currently comment only on*

*the role of the giant CCN only for Nd >100.  Using larger bins has not made the region less noisy.*

-I'm not sure I follow the reasoning on line 166 starting with 'Consequently'. Can you elaborate on this and further tie this to the prior two sentences?

*The entire paragraph has been rewritten and linked to the last sentence starting 'Consequently…'. (L168):*

*In addition to acting as a sink for the cloud Nd through the sedimentation of droplets, precipitation plays a key role in the scavenging of CCN (wet or below-cloud scavenging), which in turn can reduce Nd. The effects of precipitation are usually seen primarily in the (upper) left quadrant (LWP > 50gm^{-2}, Nd < 50cm^-3), i.e., for clouds with a high initial LWP and a low Nd. An overall positive change is seen in the dNd field in this region for both precipitating and non-precipitating clouds (red region in figures  1a,b, which is possibly a regression to the mean effect (discussed in the next paragraph). In addition to precipitation, other processes such as the primary production of CCN from sea spray, and entrainment of aerosols from the free troposphere (especially closer to the coast) can possibly act as significant sources of Nd for clouds with an initially low Nd.*

*However, precipitation rates as low as 1mm/d have been shown to be effective in reducing Nd by a factor of three over the SE Pacific {Wood2012}. The DoRs between precipitating and not-precipitating clouds (figure 1c) reveal that precipitation acts as a sink for the cloud Nd, with a reduction of Nd observed in more strongly precipitating cases. Precipitation results in a smaller overall net increase in dNd (i.e., the change in Nd over three hours) with figure 1a showing lighter reds and darker greens.*

*There is a smaller decrease (larger increase) in dNd for not-precipitating clouds  in figure 1b (darker reds and lighter greens). Consequently, the corresponding DoR,  ~(difference between figures 1a and b, i.e., a- b , is negative (figure 1c).*

-Paragraph beginning on line 194: Can you add one additional paragraph explicitly stating how these numbers concerning the two pathways are determined. It wasn't clear on first reading how this really worked in relation to the figures.

*More details have now been added in the section (L292). There was an error stating that figure 3 was being used, when it should have been figure 2.*

-Are the figures in the appendix using the data from Eastman et al. (2019)? If so, rain rate estimates in that dataset are constructed from AMSR/E and AMSR/2, with CloudSat only used to tune the relationships. If that is the case, I would change the labels to read 'AMSR' rain rates instead of CloudSat.

*Yes, the data is from Eastman et al 2019. The label has been changed to 'AMSR rain rates'.*

Minor points:

-Nd and $r_e$ are used somewhat interchangeably in the paper. Since the figures only deal in Nd, it may be easier to interpret the work if you keep the discussion limited to one variable, but also thoroughly explain the relationship between Nd and $r_e$ in the data section.

*We have tried to restrict re to section 3.2 as this is where it is most relevant. However, we have had to include re in a few other places as we felt it was important for the corresponding discussion.*

-Why restrict the work to just the SE Atlantic? Could there be regional differences if compared to Pacific Sc decks? Or more robust results?

*We have included a new section looking at regional differences. (Section 3.3)*

-Line 192, 2 dashes: Instead of dashes, maybe label these two mechanisms as 'pathways' and reference Figure 3 here directly.

*We have changed this according to the reviewer's suggestion (L283).*

-Line 194: Contrary to which other results? This paper also shows that increased wind speed leads to decreased Nd and stronger rain rates:

Eastman, R., McCoy, I.L., Wood, R., 2022. Wind, Rain, and the Closed to Open Cell Transition in Subtropical Marine Stratocumulus. Journal of Geophysical Research: Atmospheres 127, e2022JD036795. https://doi.org/10.1029/2022JD036795

*We have removed 'contrary' and included a reference to the above (and another) work.*

*(L290)*